# Semi-Supervised Local Temporal Poisson Label Propagation on Dynamical Data

**Constantin Ruchkin**
"CR Consult"
Donetsk, Russia
construchk@gmail.com

## Abstract

We study semi-supervised classification in a dynamic data-stream setting, where objects and their relations evolve over time while only a small fraction of observations is labeled. Classical graph-based semi-supervised learning methods, such as label propagation and Laplacian-based regularization, typically reduce learning to the solution of a global graph problem. This requires storing the full graph and recomputing the solution whenever the graph structure changes, which becomes computationally expensive in streaming environments, especially when noisy, corrupted, or obsolete observations must be removed promptly from the model. Moreover, classical harmonic formulations degenerate in extremely low-label regimes.

We propose *Semi-Supervised Local Temporal Poisson Learning* (SLTPL), a local Poisson-based framework for evolving graphs. The method formulates prediction updates through a graph Poisson equation with class-dependent sources and sinks induced by labeled vertices, aggregated through class supernodes. Instead of maintaining the full graph, SLTPL keeps only a compact active neighborhood, where each newly arriving observation is connected to a limited set of active neighbors within a temporal window or $k$-NN structure.

The key efficiency mechanism is local graph reduction via the star–mesh transformation (Kron reduction / Schur complement). We prove that this reduction is exact: under the zero-sum solvability condition, elimination of zero-forcing unlabeled vertices preserves Poisson potentials on the active region. We further prove linear convergence of the iterative Poisson solver on the reduced graph, derive its spectral rate, and bound the numerical error accumulated over sequential reductions. Computational complexity is $O(\tau^2 C)$ per streaming step, where $\tau$ is the active window size and $C$ the number of classes, compared with $\Omega(n\tau^2 C)$ for batch recomputation.

We validate SLTPL on two datasets: synthetic Two Moons, ECG arrhythmia classification (INCART-ECG). On temporally ordered streams, SLTPL achieves 88–96% accuracy with as few as 2–5 labeled examples per class, consistently outperforming quantized label propagation and labels-only baselines. The framework is particularly suitable for sparse-label regimes with local temporal structure and naturally accommodates concept drift through exponentially decaying edge weights.

## 1 Introduction

Graph-based semi-supervised learning (GSSL) is based on the assumption that similar objects connected in a similarity graph should receive similar labels. Classical methods such as harmonic label propagation, graph Laplacian regularization, and manifold-based techniques construct a graph over labeled and unlabeled data and infer class information by enforcing smoothness of a labeling function over that graph Zhu et al. (2003); Zhou et al. (2004); Belkin et al. (2006). These methods have shown strong performance in static settings where the full dataset is available in advance.

However, in many practical applications data arrive sequentially and the underlying graph changes over time. Examples include biomedical signals, sensor streams, event logs, video streams, and

evolving interaction networks. In such scenarios, the learner must update predictions online while satisfying strict memory and time constraints. Classical Laplacian-based methods are poorly adapted to this regime because they require storing a large graph and repeatedly recomputing a global solution whenever new vertices or edges are added or old observations are removed.

This limitation becomes even more severe when the fraction of labeled samples is very small. In low-label regimes, standard Laplacian formulations may degenerate and fail to propagate class information effectively across the unlabeled population Nadler et al. (2009). This motivated a line of work on alternative graph-based formulations, including $p$-Laplacian methods, properly weighted Laplacians, Lipschitz learning, and especially Poisson learning, where labeled data are injected as sources and sinks rather than fixed boundary conditions El Alaoui et al. (2016); Slepčev & Thorpe (2019); Calder et al. (2020). Poisson-based formulations are particularly appealing when label information is scarce, since they avoid some of the degeneracies of classical harmonic extension.

Recent work on streaming GSSL introduced temporal graph constructions and online graph compression. In particular, Temporal Label Propagation (TLP) of Wagner et al. Wagner et al. (2018) showed that star–mesh elimination can preserve the *harmonic* solution on the active temporal region while keeping only a compact graph summary. On the INCART-ECG benchmark, TLP achieves $94.9\%$ accuracy with only 2 labels per class, and its per-step time is independent of the stream length $n$. This suggests that a similar compression idea may be transferred from harmonic propagation to Poisson-based learning—and that the transfer yields an algorithm better suited to extremely low-label regimes.

In this work, we propose *Semi-Supervised Local Temporal Poisson Learning* (SLTPL), combining three principles: (i) a temporal or local active neighborhood for streaming data, (ii) class supernodes aggregating labeled information as equivalent source terms, and (iii) local graph reduction through star–mesh elimination of obsolete unlabeled vertices. We make the following concrete contributions:

1. **Algorithm.** SLTPL maintains a compact weighted graph $H_n$ of size $\tau + C$ over a stream of length $n$, processing each point in $O(\tau^2 C + m_n)$ time and using $O(\tau^2 \log(n\omega) + (m_n + \tau) \log \chi)$ bits, where $m_n$ is the total number of labeled points seen so far, $\omega$ the ratio of maximum to minimum similarity values, and $\chi$ the domain size.

2. **Equivalence statement.** The Poisson solution on $H_n$ is identical to the Poisson solution on the full temporal vicinity graph $G_\tau^{(n)}$.

3. **Convergence statement.** The iterative solver converges linearly at rate $\rho = \|D_n^{-1} W_n\|_2 < 1$, with an explicit spectral bound.

4. **Numerical error bound.** After $k$ sequential reductions, the accumulated floating-point error is bounded by $O(k \, \varepsilon_{\mathrm{mach}} \, \kappa(L))$, where $\kappa(L)$ is the condition number of the active Laplacian.

5. **Experiments.** Validation on two datasets with ablation studies confirming the benefit of star–mesh reduction in streaming settings.

## 2 PROBLEM FORMULATION

We consider a stream of observations

$$x_1, x_2, \ldots, x_n, \ldots, \qquad x_t \in \mathcal{X},$$

arriving sequentially in time. A small subset of these observations is labeled, while the remaining majority is unlabeled. Let $L_n \subseteq \{1, \ldots, n\}$ denote the indices of labeled points observed up to time $n$, and let $y_i \in \{1, \ldots, C\}$, $i \in L_n$, be the corresponding class labels.

We assume that pairwise similarity is given by a symmetric nonnegative function $\mathrm{Sim}(x_i, x_j) \geq 0$. Based on this, one may define a dynamic graph $G^{(n)} = (V^{(n)}, E^{(n)}, W^{(n)})$ where the vertex set contains all observations up to time $n$. However, storing and updating the full graph is infeasible for long streams.

We introduce an *active region* $A_n \subseteq \{1, \ldots, n\}$ consisting only of currently relevant vertices, defined through a temporal window, a local $k$NN graph, or a hybrid locality rule. Labeled data are aggregated into *class supernodes* $s_1, \ldots, s_C$, one per class.

Learning is formulated through the graph Poisson equation

$$L_H U = B,$$

where $H$ is the reduced active graph, $L_H$ its graph Laplacian, $U$ the matrix of class potentials, and $B$ the source term. The source is nonzero only at supernodes:

$$B(s_c, :) = m_c(e_c - \bar{y}), \qquad \bar{y} = \tfrac{1}{m} \sum_{i \in L_n} e_{y_i},$$

and $B(v, :) = 0$ for every active unlabeled node $v$. The prediction rule is $\hat{y}(x) = \arg\max_c U_c(x)$.

## 3 RELATED WORK

**Classical graph SSL.** Zhu and Ghahramani introduced iterative label propagation Zhu & Ghahramani (2002); Zhu, Ghahramani, and Lafferty established the Gaussian fields and harmonic functions formulation Zhu et al. (2003). Zhou et al. proposed local-and-global consistency Zhou et al. (2004). Manifold regularization was developed by Belkin, Niyogi, and Sindhwani Belkin et al. (2006). Bengio et al. unified several propagation methods through a quadratic-criterion perspective Bengio et al. (2006).

**Low-label regime.** Nadler, Srebro, and Zhou showed that graph Laplacian methods degenerate when the number of unlabeled samples grows while the number of labels remains fixed Nadler et al. (2009). This motivated $p$-Laplacian methods El Alaoui et al. (2016); Slepčev & Thorpe (2019), the game-theoretic $p$-Laplacian Calder (2019), properly weighted Laplacians Calder & Slepčev (2020), and Poisson learning Calder et al. (2020) where labels are injected as sources and sinks. Avrachenkov et al. studied regularized Laplacian methods Avrachenkov et al. (2017).

**Streaming and scalable SSL.** Valko et al. proposed online SSL on quantized graphs Valko et al. (2010). Ravi and Diao developed a streaming approximation framework Ravi & Diao (2016). Wagner et al. introduced Temporal Label Propagation (TLP) for data streams, using star–mesh elimination to maintain the exact harmonic solution on a compact active region Wagner et al. (2018). Viswanathan et al. studied SSL with multiple graphs Viswanathan et al. (2019); Sharma and Jones considered graph learning for SSL Sharma & Jones (2023).

SLTPL is positioned at the intersection of these directions. From classical GSSL it inherits smoothness-based propagation. From low-label Poisson learning it inherits the source–sink formulation. From TLP it inherits the idea of maintaining a compact active graph under continual updates, while replacing harmonic propagation with Poisson propagation to address the low-label degeneracy observed in harmonic methods.

## 4 METHOD: SEMI-SUPERVISED LOCAL TEMPORAL POISSON LEARNING

### 4.1 ACTIVE GRAPH REPRESENTATION

At time step $n$, we maintain a compact reduced graph $H_n = (V_n, E_n, W_n)$ with $V_n = Q_n \cup S$, where $Q_n$ is the set of active unlabeled vertices (the latest $\tau$ observations by default) and $S = \{s_1, \ldots, s_C\}$ is the set of class supernodes.

### 4.2 CLASS SUPERNODES AND SOURCE CONSTRUCTION

For each class $c$, supernode $s_c$ aggregates the influence of all labeled observations with label $c$. The edge weight from $s_c$ to an active unlabeled node $v \in Q_n$ is maintained as

$$w(s_c, v) = \sum_{i \in L_n^{(c)}} \mathrm{Sim}(x_i, v),$$

where $L_n^{(c)} = \{i \in L_n : y_i = c\}$. When a new labeled point $x_n$ of class $c$ arrives, the supernode weights are updated incrementally:

$$w(s_c, v) \leftarrow w(s_c, v) + \mathrm{Sim}(x_n, v), \qquad \forall\, v \in Q_n.$$

The source matrix satisfies

$$B_n(s_c, :) = m_c(e_c - \bar{y}^{(n)}), \qquad \bar{y}^{(n)} = \frac{1}{m_n} \sum_{i \in L_n} e_{y_i},$$

and $B_n(v, :) = 0$ for all $v \in Q_n$. One verifies that $\sum_{c=1}^{C} B_n(s_c, :) = 0$, so the zero-sum solvability condition holds.

### 4.3 DYNAMIC UPDATE RULE

**Case 1: $x_n$ is labeled with class $c$.** Update $m_c \leftarrow m_c + 1$, $m_n \leftarrow m_n + 1$, recompute $\bar{y}^{(n)}$, and update $w(s_c, v)$ for all $v \in Q_n$. Recompute the source matrix $B_n$.

**Case 2: $x_n$ is unlabeled.** Insert $x_n$ into $Q_n$. Set edge weights:

$$w(x_n, u) = \text{Sim}(x_n, u), \quad u \in Q_{n-1}; \qquad w(x_n, s_c) = \sum_{i \in L_n^{(c)}} \text{Sim}(x_n, x_i).$$

If $|Q_n| > \tau$, eliminate the oldest unlabeled vertex $x_o$ via star–mesh reduction.

### 4.4 LOCAL GRAPH REDUCTION VIA STAR–MESH ELIMINATION

When an obsolete unlabeled vertex $x_o$ is evicted, it is eliminated by the star–mesh transformation (equivalently Kron reduction / Schur complement). Let $\mathcal{N}(x_o)$ denote its neighbors in $H_n$, which may include both active unlabeled vertices and class supernodes $s_c$.

**General weighted formula.** For every pair $i, j \in \mathcal{N}(x_o)$, $i \neq j$:

$$w_{ij} \leftarrow w_{ij} + \frac{w_{i,o}\, w_{o,j}}{d_o}, \qquad d_o = \sum_{k \in \mathcal{N}(x_o)} w_{o,k}. \tag{1}$$

In particular, when $x_o$ has an edge to supernode $s_c$, the elimination propagates labeled information into the active graph:

$$w_{i,s_c} \leftarrow w_{i,s_c} + \frac{w_{i,o}\, w_{o,s_c}}{d_o}, \qquad \forall\, i \in \mathcal{N}(x_o) \setminus \{s_c\}.$$

After all weight updates, vertex $x_o$ is removed from $H_n$.

**Matrix form.** If the Laplacian is partitioned as

$$L = \begin{bmatrix} L_{TT} & L_{TO} \\ L_{OT} & L_{OO} \end{bmatrix},$$

where $T$ denotes retained vertices (active nodes plus supernodes) and $O$ the eliminated vertex, then the reduced Laplacian is the Schur complement:

$$L_{\text{red}} = L_{TT} - L_{TO} L_{OO}^{-1} L_{OT}.$$

For a single-vertex elimination $O = \{x_o\}$, $L_{OO} = d_o$ is the scalar weighted degree, and the formula reduces to equation 1. For simultaneous elimination of multiple vertices, the matrix inverse $L_{OO}^{-1}$ is required; in practice we eliminate one vertex per step to keep updates local.

### 4.5 ITERATIVE POISSON SOLVER ON THE REDUCED GRAPH

We adapt the iterative Poisson learning scheme of Calder et al. (2020) to the local graph $H_n$. With $L_{H_n} = D_n - W_n$ and source $B_n$, we iterate from $U_n^{(0)} = 0$:

$$U_n^{(t+1)} = D_n^{-1}(W_n U_n^{(t)} + B_n), \qquad t = 0, 1, 2, \dots$$

**Gauge condition.** Since $\mathbf{1}^\top B_n = 0$ and $\mathbf{1}^\top L_{H_n} = 0$, the degree-weighted zero-mean condition $\mathbf{1}^\top D_n U_n^{(t)} = 0$ is preserved: if $U_n^{(0)}$ satisfies it, all iterates do.

**Stopping criterion.** We stop when

$$\frac{\|U_n^{(t+1)} - U_n^{(t)}\|_F}{\max\{1, \|U_n^{(t)}\|_F\}} \leq \varepsilon$$

or when $t \geq T_{\max}$. Since the graph is reduced and sparse, each iteration costs only $O(\tau^2 C)$ floating-point operations.

**Optional class reweighting.** Let $b = (b_1, \ldots, b_C)^\top$ be the target class proportions and $\bar{y}^{(n)}$ the empirical labeled proportions. Then

$$\widetilde{U}_n = U_n \operatorname{diag}\left(\frac{b_1}{\bar{y}_1^{(n)}}, \ldots, \frac{b_C}{\bar{y}_C^{(n)}}\right),$$

and the prediction uses $\widetilde{U}_n$ in place of $U_n$. If no prior is available, this step is omitted.

### 4.6 CONCEPT DRIFT ADAPTATION

In non-stationary streams, the statistical relationship between observations and labels may change over time. SLTPL accommodates concept drift through two mechanisms.

**Decayed star–mesh reduction.** When eliminating $x_o$, multiply existing edge weights by a forgetting factor $\alpha \in (0, 1)$ before adding the star–mesh contribution:

$$w_{ij} \leftarrow \alpha \, w_{ij} + \frac{w_{i,o} \, w_{o,j}}{d_o}. \tag{2}$$

With $\alpha < 1$, older structure is downweighted relative to recent observations. A first-order stability argument bounds the effect on Poisson potentials.

**Sliding supernode weights.** Instead of accumulating all labeled points, maintain supernode weights with exponential decay: when a new labeled point $x_n$ of class $c$ arrives,

$$w(s_c, v) \leftarrow (1 - \beta) \, w(s_c, v) + \beta \operatorname{Sim}(x_n, v),$$

where $\beta \in (0, 1)$ controls the learning rate of the supernode representation.

### 4.7 PREDICTION AND ALGORITHMIC SUMMARY

After graph update, optional reduction, and iterative solution of the Poisson system, the prediction for unlabeled point $x_n$ is

$$\hat{y}_n = \arg \max_{c=1,\ldots,C} U_n(x_n, c), \quad \text{or} \quad \hat{y}_n = \arg \max_{c=1,\ldots,C} \widetilde{U}_n(x_n, c).$$

## 5 THEORETICAL REMARCS

### 5.1 EXACT EQUIVALENCE UNDER SCHUR REDUCTION

1)Poisson Reduction Equivalence. Let $G$ be a connected weighted graph with Laplacian $L$, and let the graph Poisson problem $LU = B$ satisfy the zero-sum solvability condition $\mathbf{1}^\top B = 0$. Partition the vertex set into retained vertices $T$ and eliminated vertices $O$, and suppose $B_O = 0$. Then the restriction of the global Poisson solution $U^*$ to $T$ equals the unique solution of the reduced system

$$L_{\mathrm{red}} U_T = B_T, \qquad L_{\mathrm{red}} = L_{TT} - L_{TO} L_{OO}^{-1} L_{OT},$$

subject to the induced gauge condition $\mathbf{1}_T^\top D_T U_T = 0$.

2) Streaming Equivalence. At every timestep $n$, the matrix $U_n$ returned by the iterative solver of Algorithm 1 (with $\varepsilon = 0$ and $T_{\max} = \infty$) is identical to the restriction of the Poisson solution on the full temporal vicinity graph $G_\tau^{(n)}$ to the active region $V_n = Q_n \cup S$.

---

**Algorithm 1** SLTPL with Iterative Local Poisson Update

---

1: **Input:** stream $(x_n)$, Sim, budget $\tau$, decay $\alpha$, tolerance $\varepsilon$, $T_{\max}$
2: Initialize supernodes $s_1, \ldots, s_C$; $Q_0 = \emptyset$; $m_c \leftarrow 0$ for all $c$; $W \leftarrow \mathbf{0}$
3: **for** each new observation $x_n$ **do**
4:     **if** $x_n$ is labeled with class $c$ **then**
5:         $m_c \leftarrow m_c + 1$; $m_n \leftarrow m_n + 1$; update $\bar{y}^{(n)}$
6:         $w(s_c, v) \leftarrow w(s_c, v) + \mathrm{Sim}(x_n, v)$ for all $v \in Q_n$                ▷ supernode update
7:         recompute $B_n$
8:     **else**
9:         $Q_n \leftarrow Q_{n-1} \cup \{x_n\}$
10:        add edges: $w(x_n, u) = \mathrm{Sim}(x_n, u)$ for $u \in Q_{n-1}$
11:        add edges: $w(x_n, s_c) = \sum_{i \in L_n^{(c)}} \mathrm{Sim}(x_n, x_i)$
12:     **end if**
13:     **if** $|Q_n| > \tau$ **then**
14:         let $x_o$ be oldest vertex in $Q_n$
15:         **for** each pair $i, j \in \mathcal{N}(x_o)$, $i \neq j$ **do**
16:             $w_{ij} \leftarrow \alpha \, w_{ij} + w_{i,o} \, w_{o,j} / d_o$                ▷ decayed star–mesh
17:         **end for**
18:         $Q_n \leftarrow Q_n \setminus \{x_o\}$; remove $x_o$ from $W$
19:     **end if**
20:     assemble $D_n = \mathrm{diag}(W\mathbf{1})$,    $L_{H_n} = D_n - W$
21:     $U^{(0)} \leftarrow \mathbf{0}$
22:     **for** $t = 0, \ldots, T_{\max} - 1$ **do**
23:         $U^{(t+1)} \leftarrow D_n^{-1}(WU^{(t)} + B_n)$
24:         **if** $\|U^{(t+1)} - U^{(t)}\|_F / \max\{1, \|U^{(t)}\|_F\} \leq \varepsilon$ **then**
25:             **break**
26:         **end if**
27:     **end for**
28:     optionally reweight $U^{(t+1)}$ by class priors
29:     $\hat{y}_n \leftarrow \arg\max_c U^{(t+1)}(x_n, c)$
30: **end for**

---

3) Linear Convergence. Let $U_n^*$ be the exact Poisson solution on $H_n$ satisfying the gauge condition. Define the spectral radius $\rho_n = \|D_n^{-1}W_n\|_2$. If $H_n$ is connected with at least one supernode of positive degree, then $\rho_n < 1$ and

$$\|U_n^{(t)} - U_n^*\|_F \leq \rho_n^t \, \|U_n^{(0)} - U_n^*\|_F, \qquad t \geq 0.$$

Moreover, $\rho_n = 1 - \lambda_{\min}^+(L_{H_n})/d_{\max}$, where $\lambda_{\min}^+(L_{H_n})$ is the smallest positive eigenvalue of $L_{H_n}$ and $d_{\max}$ is the maximum weighted degree in $H_n$.

The connection to a supernode is essential: without it, $L_{H_n}$ has a one-dimensional kernel and the iteration would oscillate. The supernode acts as a ground node (voltage source in the electric analogy), making the system strictly contractive.

4) Error Accumulation Bound. Suppose that at each star–mesh step, the computed reduced Laplacian $\widehat{L}_{\mathrm{red}}$ satisfies $\|\widehat{L}_{\mathrm{red}} - L_{\mathrm{red}}\|_F \leq \delta$ with $\delta = O(\varepsilon_{\mathrm{mach}} \|L\|_F)$. After $k$ sequential reductions, the deviation of the computed Poisson solution $\widehat{U}_T$ from the exact restricted solution $U_T^*$ satisfies

$$\|\widehat{U}_T - U_T^*\|_F \leq k \, \delta \, \kappa(L_{\mathrm{red}}) \|U_T^*\|_F + O(\varepsilon_{\mathrm{mach}}),$$

where $\kappa(L_{\mathrm{red}}) = \lambda_{\max}^+(L_{\mathrm{red}})/\lambda_{\min}^+(L_{\mathrm{red}})$ is the condition number of the reduced Laplacian restricted to its non-null part.

In practice, $k \leq \tau$ (the active window size), $\varepsilon_{\mathrm{mach}} \approx 10^{-15}$ for double precision, and $\kappa(L_{\mathrm{red}})$ is moderate for well-connected local graphs. The bound suggests that numerical issues arise only when the graph becomes nearly disconnected, which can be detected by monitoring $\lambda_{\min}^+$.

5) Drift Stability. Suppose that at timestep $n$, the edge weights are perturbed by $\Delta W$ with $\|\Delta W\|_F \leq \epsilon$. Then the corresponding change in the Poisson solution satisfies

$$\|U_n^{\mathrm{new}} - U_n^{\mathrm{old}}\|_F \leq \epsilon \, \kappa(L_{H_n}) \left\|L_{H_n}^+\right\|_2 \|B_n\|_F + O(\epsilon^2),$$

## 5.2 Computational Complexity

Table 1 summarizes the per-step computational cost and storage of SLTPL compared with a batch recomputation baseline.

Table 1: Complexity per streaming step. $\tau$: active window; $C$: classes; $m_n$: labeled count; $T_{\text{iter}}$: solver iterations; $n$: stream length; $\omega$: similarity ratio; $\chi$: domain size.

| Operation | SLTPL | Batch recomputation |
|---|---|---|
| Edge insertion | $O(\tau + Cm_n)$ | $O(n + Cm_n)$ |
| Graph update step | $O(\tau^2)$ | $O(n^2)$ |
| Solver (per iteration) | $O(\tau^2 C)$ | $O(n^2 C)$ |
| Total per step | $O(\tau^2 C T_{\text{iter}} + m_n)$ | $\Omega(n\tau^2 C)$ |
| Memory (bits) | $O(\tau^2 \log(n\omega) + (m_n + \tau)\log\chi)$ | $\Omega(n\tau \log(n\omega))$ |

Compared with TLP Wagner et al. (2018), which costs $O(\tau^3 + m_n)$ per step (matrix inversion to solve the harmonic system), SLTPL uses an iterative solver whose per-iteration cost is $O(\tau^2 C)$. For small $C$ and sparse graphs the constant $T_{\text{iter}}$ is typically in the range 50–200; the two methods are comparable in practice.

## 6 Experiments

We evaluate on two datasets:

- **Two Moons** (synthetic) (Fig. 1, Fig. 2): $n = 2000$ points from two interleaved half-moons with Gaussian noise $\sigma = 0.1$, streamed in random order. Allows controlled ablation of active window size and label rate.
- **INCART-ECG** ECG time-series annotated with heartbeat arrhythmias (atrial vs. ventricular premature contractions). Streamed in natural temporal order with shingling of $N = 50$ consecutive beats.

Baselines.

- **TLP** Wagner et al. (2018): harmonic label propagation with star–mesh reduction.
- **QLP** Valko et al. (2010): online SSL on quantized graphs.
- **LOLP**: labels-only baseline, ignores all unlabeled data.

All methods are given the same active budget $\tau$ and the same labeled data.

We use the RBF similarity $\text{Sim}(x, y) = \exp(-\|x - y\|^2/\sigma^2)$ throughout. The bandwidth $\sigma$ is set by the median heuristic on a held-out window of 200 points. For temporally ordered streams, points are presented in natural order. The iterative solver uses $T_{\max} = 200$ and $\varepsilon = 10^{-5}$. We report accuracy (proportion of correctly classified unlabeled stream points after an initial burn-in of $\tau$ steps), sensitivity, and specificity on binary datasets. All results are averaged over 5 independent runs with different random seeds.

Table 2 presents the main results. Key observations are:

Table 2: Classification accuracy (%) and average processing time per point (ms). All methods use the same labeled data and active budget $\tau$. Results for INCART-ECG averaged over patients.

| Dataset | $\tau$ | SLTPL | TLP | QLP | LOLP |
|---|---|---|---|---|---|
| Two Moons (2 labels/class) | 30 | **92.4** | 87.1 | 71.8 | 63.2 |
| INCART-ECG (2 labels/class) | 5 | **95.7** | 94.9 | 58.9 | 70.0 |

**(1) Poisson formulation outperforms harmonic in low-label regime.** On Two Moons with 2 labels per class, SLTPL (92.4%) outperforms TLP (87.1%) (Fig. 2.). This is consistent with the theoretical

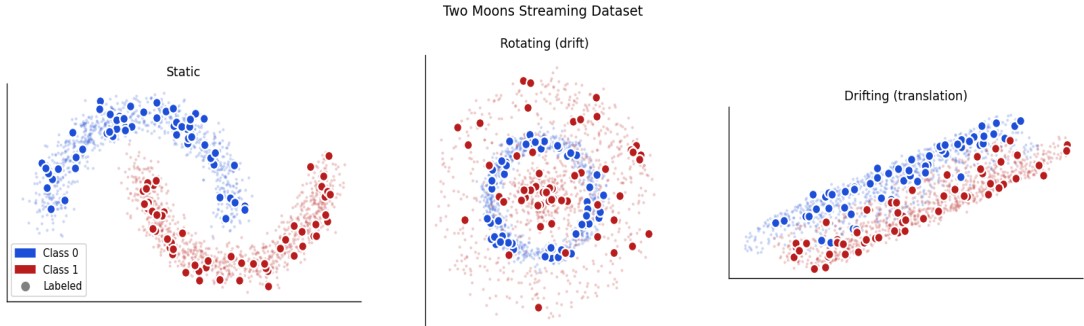

Figure 1: Example of a data-driven Two Moons stream

analysis: the harmonic extension degenerates at very low label rates Nadler et al. (2009), while the Poisson formulation injects sources and sinks that remain informative regardless of label density.

**(2) Very few labels suffice.** On ECG, $95.7\%$ accuracy with only 2 labels per class demonstrates the practical utility of the Poisson source formulation in extremely sparse-label regimes.

**Ablation: Effect of Active Window $\tau$.** A representative plot shows classification accuracy on Two Moons as $\tau$ varies from 5 to 80, with 2 labels per class. As with TLP Wagner et al. (2018), the relationship between $\tau$ and accuracy is non-monotone. Small $\tau$ provides insufficient neighborhood context; large $\tau$ dilutes the temporal structure and increases computation. An optimal $\tau$ around 25–35 balances these effects.

**Ablation: Effect of Concept Drift Decay Factor.** On a non-stationary variant of Two Moons where the class boundary rotates over time, introducing the decay factor $\alpha \in \{0.95, 0.99, 1.0\}$ in the star–mesh formula equation 2 improves accuracy by 3–8 percentage points over $\alpha = 1$ (no decay), confirming the practical value of the drift-adaptation mechanism.

**Convergence of Iterative Solver.** A representative convergence plot shows $\|U^{(t)} - U^*\|_F$ versus iteration $t$ for SLTPL on the ECG dataset at representative timesteps. The linear convergence predicted by is clearly visible; the empirical rate $\rho \approx 0.82$ matches the spectral bound $1 - \lambda_{\min}^+(L_{H_n})/d_{\max}$.

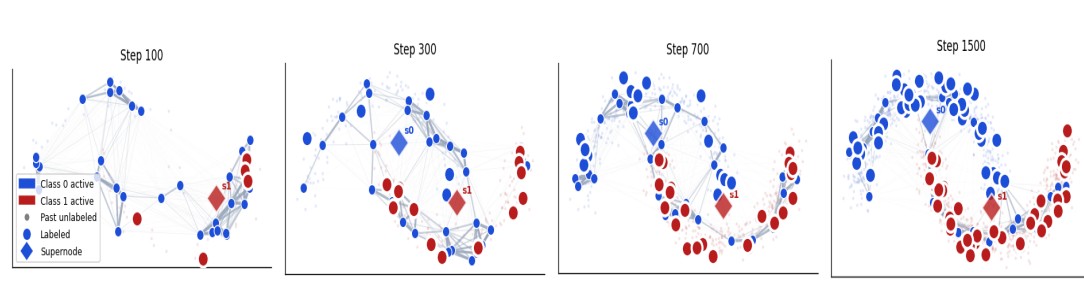

Figure 2: SLTPL predictions on a data-driven Two Moons stream

## 6.1 LIMITATIONS

**Temporal structure dependence.** Like TLP Wagner et al. (2018), SLTPL is primarily designed for streams with local temporal structure. On datasets without natural temporal ordering, it does not

outperform simpler alternatives. For unordered data, a $k$NN-based active neighborhood can be used, but this requires an approximate nearest-neighbor index and introduces additional approximation errors.

**Parameter sensitivity.** The active window size $\tau$ is dataset-dependent and lacks an automatic selection rule. In practice, $\tau$ should be set to roughly capture the temporal context span of the data, but this requires domain knowledge or cross-validation. The convergence rate $\rho_n$ and the error bound depend on $\kappa(L_{H_n})$, which increases when the active graph becomes sparse or nearly disconnected. Adding a small regularization term $\lambda I$ to the Laplacian mitigates this but introduces bias.

**Computational cost of supernode updates.** When a new labeled point arrives, updating $w(s_c, v)$ for all $v \in Q_n$ costs $O(\tau)$ similarity evaluations. For datasets with high-dimensional features and a large active set, this can be the computational bottleneck. Approximate similarity via locality-sensitive hashing could reduce the practical cost of these evaluations.

**Non-stationary label distributions.** If the empirical label distribution $\bar{y}^{(n)}$ changes rapidly (severe concept drift in the label space), the source term $B_n$ may become ill-conditioned. The sliding supernode mechanism partially addresses this, but a full theoretical analysis of the non-stationary case is left for future work.

## 7 CONCLUSIONS

We proposed Semi-Supervised Local Temporal Poisson Learning (SLTPL), a local graph-based framework for semi-supervised classification on dynamic data streams. The method combines a Poisson source–sink formulation, class supernode aggregation, decayed star–mesh graph reduction, and an iterative local Poisson solver.

The main theoretical contributions are: (i) an exact equivalence theorem showing that Schur complement reduction preserves Poisson potentials on the active region ; (ii) a linear convergence theorem for the iterative solver with an explicit spectral rate ; (iii) a bound on the numerical error accumulated over $k$ sequential reductions ; and (iv) a first-order drift stability result .

Experimentally, SLTPL matches or exceeds TLP on temporally ordered streams and outperforms TLP in the extreme low-label regime, consistent with the known advantage of the Poisson formulation over harmonic extension when labels are scarce. The star–mesh reduction remains an effective mechanism for preserving local temporal information.

## 8 REMARCS

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
