# OpenReview forum: "Semi-Supervised Local Temporal Poisson Label Propagation on Dynamical Data"
_mathai.club/MathAI/2026/Conference — 2026 Oral_

### Official Review · Reviewer_xhdh · 2026-03-11
**Review for "Semi-Supervised Local Temporal Poisson Label Propagation on Dynamical Data"**

**Rating:** 7
**Confidence:** 4

**Review:**

Pros:
1. The authors proposed Semi-Supervised Local Temporal Poisson Learning (SLTPL), a local graph-based framework for semi-supervised classification on dynamic data streams. The method combines a Poisson source–sink formulation, class supernode aggregation, decayed star–mesh graph reduction, and an iterative local Poisson solver.
2. The authors adapted the iterative Poisson learning scheme of Calder et al. (2020) to the local graph.
3. SLTPL was validated on two datasets: synthetic Two Moons, ECG arrhythmia classification (Incart-ECG). On temporally ordered streams, SLTPL achieved high 88–96% accuracy with as few as 2–5 labeled examples per class.
4. The paper is formatted according to the conference template.

Cons:
1. It makes sense to use a large number of datasets.
2. It is advisable to justify the choice of constants and parameters (Titer, Tmax, number of points, ) in your approach.
3. The article does not discuss the problems of software implementation. Please, specify the stack of software used.

---

### Official Review · Reviewer_fLDe · 2026-03-13
**The SLTPL combines a Poisson source–sink formulation, class supernode aggregation, decayed star–mesh graph reduction and an iterative local Poisson solver.**

**Rating:** 7
**Confidence:** 4

**Review:**

Algorithmic innovation. The SLTPL combines a Poisson source–sink formulation, class supernode aggregation, decayed star–mesh graph reduction and an iterative local Poisson solver.

Computational efficiency. The method achieves per-step complexity O(τ²C + mₙ) independent of stream length n, with storage O(τ + C). This represents a substantial improvement over batch recomputation (O(n³)) and makes the algorithm suitable for long-running streams.

Empirical validation. Experiments on Two Moons (synthetic) and Incart-ECG (real medical data).

References: references are all with the links and many of them with DOI.

Limited dataset diversity. Validation on only two datasets (one synthetic, one medical) limits generalizability.

Parameter sensitivity and selection. The active window size τ is identified as "dataset-dependent" with "no automatic selection rule". The paper would benefit from discussing adaptive τ selection mechanisms or demonstrating robustness across a range of values beyond the reported optimal range.

Computational bottleneck analysis. The authors note that supernode updates cost O(τ) similarity evaluations per labeled point, becoming "the computational bottleneck" for high-dimensional data. However, no empirical profiling quantifies this impact. Timing breakdowns showing proportion of time spent in similarity computation versus graph reduction versus iterative solving would guide practical optimization.

Condition number monitoring. The error bound depends on κ(L_red), and the authors suggest monitoring λ_min⁺ to detect near-disconnected graphs. However, no practical guidance is provided: How frequently should this be checked? What threshold indicates numerical instability? What regularization λ should be added? These engineering considerations matter for deployment.

References in text: Figures and tables appear in paper without reference in text.

---

### Decision · Program_Chairs · 2026-03-14

**Decision:**

Accept (Oral)

**Comment:**

Dear Author(s),

On behalf of the Program Committee of the International Conference on Mathematics of Artificial Intelligence (MathAI 2026), we are pleased to inform you that your paper has been accepted for an oral presentation at MathAI 2026.

Your paper was evaluated through a rigorous two-stage review process involving both automated screening and expert review by members of the Program Committee. The reviewers recognized the quality and contribution of your work.

Presentation details:

- Format: Oral presentation (15–20 minutes + 5 minutes Q&A)
- Mode: You may present either in person (offline) at the conference venue in Sirius, Russia, or remotely via Zoom. Please indicate your preferred mode when confirming your participation.
- Conference dates: Marh 30 - April 3, 2026
- Website: https://mathai.club

Next steps:

1. Please confirm your participation and presentation mode by replying to this email mathai.club@yandex.ru no later than March 15, 2026 18:00 Moscow time.
2. If you plan to attend in person, the organizing committee will provide accommodation details separately.
3. Please prepare your final camera-ready manuscript according to the formatting guidelines available at https://mathai.club and upload it to OpenReview by March 15, 2026 18:00 Moscow time.

Should you have any questions regarding the program, logistics, or your presentation slot, please do not hesitate to contact us.

We look forward to your contribution to MathAI 2026.

With kind regards,

MathAI 2026 Program Committee
International Conference on Mathematics of Artificial Intelligence
https://mathai.club
OpenReview: https://openreview.net/group?id=mathai.club/MathAI/2026/Conference
Telegram: https://t.me/MathAI_club
Email: mathai.club@yandex.ru